# High Red–Blue Light Ratio Promotes Accelerated In Vitro Flowering and Seed-Set Development in *Amaranthus hypochondriacus* Under a Long-Day Photoperiod

**DOI:** 10.3390/plants14203134

**Published:** 2025-10-11

**Authors:** Alex R. Bermudez-Valle, Norma A. Martínez-Gallardo, Eliana Valencia-Lozano, John P. Délano-Frier

**Affiliations:** 1Departamento de Biotecnología y Bioquímica, Centro de Investigación y de Estudios Avanzados del IPN, Unidad Irapuato, Irapuato 36824, Guanajuato, Mexico; alex.bermude@cinvestav.mx (A.R.B.-V.); norma.martinez@cinvestav.mx (N.A.M.-G.); 2Laboratorio de Ciencias Agrogenómicas, Universidad Nacional Autónoma de México ENES Campus León, León de los Aldama 37684, Guanajuato, Mexico; evalencial@enes.unam.mx

**Keywords:** flowering-related genes, grain amaranth, in vitro cultivation, long-day photoperiods, red–blue light

## Abstract

Grain amaranths are recalcitrant to conventional in vitro plant regeneration by organogenesis de novo or through somatic embryogenesis. Consequently, floral organogenesis by these methods, representing the culminating developmental point in angiosperms, is rarely achieved. In the present study, the manipulation of in vitro flowering was explored as part of a strategy designed to overcome grain amaranth’s regeneration recalcitrance. It led to an efficient and reproducible in vitro protocol in which half-longitudinally dissected zygotic embryos generated fully developed *Amaranthus hypochondriacus* (*Ah*) plants. The use of high-irradiance illumination with LED lamps with a 3:1 red–blue irradiance ratio was a critical factor, leading to a 70% rate of early flowering events under flowering-inhibiting long-day photoperiod conditions. Contrariwise, no flowering was induced under LED white lights. All in vitro flowering *Ah* plants yielded viable seeds. To understand the basic molecular mechanisms of the phenomenon observed, gene expression patterns and principal component analysis of key flowering-related genes were analyzed after cultivation in vitro for 4, 8, and 12 weeks under both lighting regimes. These coded for photoreceptors, photomorphogenetic regulators, embryogenic modulators, and flowering activators/repressors. The results highlighted the upregulation of key flowering-regulatory genes, including *CONSTANS*, *FLOWERING LOCUS T*, and *LEAFY*, together with the downregulation of the floral repressor *TERMINAL FLOWER1*. Ribosome biogenesis- and seed-development-related genes were also differentially expressed, supporting a key role in this process for protein synthesis and embryogenesis. A model is proposed to explain how this light-regulated molecular framework enables in vitro flowering and seed production in *Ah* plants kept under long-day photoperiods.

## 1. Introduction

Grain amaranths (*Amaranthus hypochondriacus* [*Ah*], *A. cruentus*, and *A. caudatus*) are ancient, highly nutritional, and drought-tolerant crops [1,2]. They are native to the American continent and are greatly valued for their, protein-rich seeds that have added nutraceutical properties, although their leaves may also be used as a vitamin-rich vegetable source [3,4,5,6]. Despite all the advantages provided by grain amaranths, there is not sufficient integration of these plants into large-scale agricultural production. An important contributing factor is the difficulty of employing mechanized methods in the grain amaranth production chain. This limitation is caused by several agronomic characteristics, such as small seed size, irregular seed maturation, and plant architecture, which complicate conventional mechanized harvesting and processing. Additionally, the lack of suitable equipment adapted for grain amaranth cultivation further restricts mechanized production. To solve these agronomical limitations, the introduction of favorable changes via genetic transformation and genome editing appears to be an attractive alternative, similar to numerous other related strategies that have had a demonstrated positive impact on several other economically important crops. Unfortunately, grain amaranths are known to be notoriously recalcitrant to plant transformation, mostly because of the difficulty in achieving plant regeneration in vitro using conventional methods, usually based on modifications of the auxin/cytokinin ratios to promote organogenesis de novo and/or somatic embryogenesis (SE) [7,8,9].

In vitro flowering is an alternative tool for plant micropropagation designed to release new cultivars into the market with greater celerity. It involves the change from the apical meristematic region into a floral meristem, where the flowers will be produced. Until now, more than 100 different plant species have been reported to produce flowers in vitro [10]. The most important factors involved in flowering are the photoperiod, light quality and intensity, temperature, and growth regulators. Grain amaranths are predominantly short-day flowering crops, except for *A. cruentus*, generally known to be photoperiod-insensitive. Therefore, days shorter than 12 h will accelerate flowering in most grain amaranth species, while in *A. cruentus*, long days usually promote robust vegetative growth [2,11,12,13,14,15,16]. Contrary to their recalcitrance to shoot–root and SE plant regeneration, amaranth plants readily undergo floral development under in vitro conditions. Tisserat and Galletta [17] found that five *Amaranthus* species, i.e., *A. caudatus*, *A. gangeticus*, *Ah*, *A. retroflexus*, and *A. viridis* exhibit prolific flowering in vitro (e.g., 80% in *A gangeticus*, 30% in *Ah*). The inflorescences could be maintained separately for several months, whereas the life cycle could be completed under tissue culture conditions within 24 to 32 weeks. Later, the use of controlled cultivation conditions based on gradual day length manipulation was able to reduce flowering and generation times to 4 and 8 weeks, respectively [18].

Plants have developed a complex mechanism to sense light and use it, together with the circadian rhythm, to distinguish the proper moment to start the flowering process [19,20,21,22,23,24]. There are several signaling pathways that regulate a plant’s transition from vegetative growth to the reproductive phase. Among the most important is the photoperiod signaling pathway, which allows plants to respond to day length. Under long-day conditions, the CONSTANS (CO) protein accumulates in leaves and activates the expression of FLOWERING LOCUS T (FT), which acts as a mobile signal, or florigen, that is transported to the apical meristem to initiate flowering [25]. The light quality is perceived by the PHYTOCHROME B (PHYB; red), PHYTOCHROME A (PHYA; far red), and CRYPTOCHROME2 (CRY2; blue) photoreceptors. The CRY2 and PHYB photoreceptors are also known to be involved in the photoperiodic control of many of the plant development processes [26,27]. These further interact with proteins such as CONSTITUTIVELY PHOTOMORPHOGENIC 1 (COP1) and SUPPRESSOR OF PHYA1 (SPA1) to modulate CO stability and influence FT expression [28]. In addition, PHYTOCHROME-DEPENDENT LATE-FLOWERING (PHL) physically interacts with CO to suppress its degradation by PHYB [29].

This work describes the induction of flowering in *Ah* plants grown in vitro using 3:1 red–blue LED luminaries (R-BLL) under a long-day photoperiod. The plants developed rapidly and successfully completed their reproductive cycle, generating flowers and viable seeds that germinated normally. Subsequently, a STRING database v12.0 network was queried to select the genes whose expression patterns could test the working hypothesis stating that enriched red light illumination accelerated flowering in *Ah* under restrictive long-day photoperiod conditions by altering the regulatory mechanisms of *CONSTANS* and *FLOWERING LOCUS T*. This information was integrated into a model in which the observed increase in CO and FT transcripts under red–blue LED conditions in a long-day photoperiod suggested a possible alternative or modified flowering-regulatory mechanism in *Ah*, distinct from the classical *A. thaliana* model. Still, the proposed mechanism remains hypothetical and will require further experimentation to be validated.

## 2. Results

### 2.1. Flowering Induction

In vitro flowering of *Ah plantlets* was induced from zygotic embryos dissected longitudinally and cultured in MS medium with 3% sucrose and 3 g/L gelrite under R-BLL. Non-inductive light conditions were implemented using WLL. Long-day (16 h light/8 h dark) photoperiods within a 18 °C (dark) to 28 °C (light) temperature range were applied in both illumination regimes. Under the R-BLL experimental conditions, which were replicated five times, 90% of the cultured embryos developed into mature plants, 71% of which successfully flowered and yielded viable seeds (an average of 139 seeds per plant), which had a mean 70% germination rate (Figure 1; Table 1). Seed setting started in flowering plants 12 weeks after the experiments were started.

### 2.2. Selection of Relevant Ah Flowering-Related Genes for Analysis

A protein–protein interaction (PPI) network using STRING-based bioinformatic data with a 0.90 confidence level was constructed using information garnered from the *A. thaliana* genome (Figure 2). This exercise yielded a protein–protein interaction circuitry that was utilized to guide the selection of genes to be quantitatively analyzed during the early flowering process observed in in vitro *Ah* plants grown under R-BLL in long-day photoperiod conditions. This experimental strategy was implemented with the aim of identifying a possible signaling pathway able to explain the role of R-BLL light in the induction of the uncharacteristic long-day flowering trait observed in short-day flowering *Ah* plants.

### 2.3. Gene Expression Analysis

The expression of homologous flowering-related genes from *Ah* plants, the selection of which was based on the exercise described above, was analyzed at 4 weeks after cultivation (WAC), when plants were in a robust vegetative stage, at 8 WAC, when the first apical flower buds appeared, and at 12 WAC, when diverse flower maturation stages were present due to the asynchronous reproductive maturation that is characteristic of *Ah* plants. The relative expression values of these genes, which were obtained under R-BLL conditions, were compared with those obtained from those *Ah* plants cultivated under WLL.

The R-BLL-to-WLL relative expression value ratios obtained during the different development periods are shown in Figure 3. In the first 4 weeks (Figure 3a), *TOC*/*PRR1*, *TFL1*, and *RPS18* were among the genes most highly induced in the leaves and stem, whereas the former two were among the only genes expressed in the meristems. This showed a prevalence of factors that sustained the repression of flower development in the meristems at 4 WAC. Subsequently, in week 8 (Figure 3b), the *FT* gene, coding for the master regulator of flowering that promotes the transition from vegetative to reproductive development in plants [30], increased to unexpectedly high expression levels under R-BLL, particularly in the meristems. Other genes that reverted to positive R-BLL to WLL relative expression ratios in the SAM/FM at 8 WAC were *SPA1*, *LFY*, *GI*, *RSP18*, *NOB1*, *MDN1*, *LEC1*, and *PHL*, while others such as *RPL21C* and, importantly, the *TFL1* flowering-repression gene, showed reduced or negative expression ratios. This expression pattern was already indicative of a molecular organization positively orientated toward the induction of flowering and seed production, considering, for example, the roles of *SPA1*, *PHL*, *NOB1*, *MDL1*, and *LEC1* in the control of photoperiodic flowering and the circadian clock, the promotion of flowering under red light, and seed, embryo, gametophyte, and pollen development, respectively [31,32,33]. Lastly, at 12 WAC (Figure 3c), the most telling changes in gene expression observed in flowers were the induction of *CO*, the central controller of the photoperiod-sensing mechanism in plants that determines flowering by regulating FT, and of the *PHYB* and *CRY2* photoreceptor genes. These could have possibly permitted the integration of upstream FT-activating signals transmitted by COP1 [34]. The R-BLL-to-WLL relative expression value ratios of *LFY* in flowers also markedly increased.

### 2.4. PCA of the Gene Expression Levels Produced Under R-BLL and a Long-Day Photoperiod at 12 WAC

Component 1 of the PCA in the 12 WAC flowering stage (*x*-axis, increasing average expression) indicated that 63.98% of the genes analyzed had significantly modified levels of expression. Here, the *FT* gene exhibited the higher contribution to the detected changes in gene expression (i.e., 90.41), making it, by far, the most relevant gene in flowering development under R-BLL conditions. It was followed by *LFY* (1.33), *CO* (0.86), *CRY2* (0.73), *SPA1* (0.70), *LEC1* (0.70), *MDN1* (0.68), *PHYB* (0.67), *RPL21C* (0.63), *NOB1* (0.58), *COP1* (0.50), *PRR1* (0.49), *PHL* (0.49), *GI* (0.47), *TFL1* (0.35), and *RPS18* (0.34) (Figure 4a). Component 2 (PCA2) (*y*-axis, increasing positive trend) explained 33.61% of the variance. The most relevant gene recognized at PCA2 during flowering at 12 WAC was *TFL1* (44.46). It was followed by *RPL18* (17.00), *LFY* (9.8), *PRR1* (7.97), *GI* (5.9), *CRY2* (5.86), *CO* (4.13), *LEC1* (2.27), *PHYB* (1.34), *PHL* (0.42), *NOB1* (0.30), *MDN1* (0.22), *RPL21C* (0.14), *FT* (0.12), *SPA1* (0.016), and *COP1* (0.001) (Figure 4a). Components 1 and 3 were constituted in ca. equal measures by genes predominantly expressed during the 8 and 12 WAC sampling time points. These corresponded to the initiation of flowering and active flower development stages, respectively, while component 2 was constituted by genes whose maximum expression levels were detected at 4 WAC, when vegetative growth was still prevalent, and flowering had not yet started (Figure 4b). The PCA results obtained from the gene expression assays in the stems and in leaves are shown as Appendix A.

### 2.5. Analysis of the Cis-Acting Elements in the Ah FT Gene

PLANTCARE was used to analyze the 2000 pb upstream promoter sequence of the *Ah FT* gene and showed that the promoter region of the *FT* gene contained 21 light-responsive elements (LREs), implying that they may have contributed to the R-BLL activation of this gene under R-B enriched light and consequently to the induction of flowering under long-day photoperiod conditions (Figure 5).

## 3. Discussion

The differentiation of the shoot apical meristem into a floral meristem promoted by FT is a unique feature of the evolution of Gymnosperms and Angiosperms [36]. It is highly relevant to agriculture because of its obvious implications for sexual reproduction and plant productivity. Although flowering development under in vitro conditions has been demonstrated in more than 100 plant species [10], the molecular mechanisms responsible for the orchestration of such a significant event under these particular growing conditions have not been analyzed in depth, particularly under controlled light regimes and in underutilized crops like *Ah* [37,38].

The present study describes the induced flowering of *Ah* under in vitro conditions in which plants were continuously exposed to R-BLL under a non-flowering-promoting long-day photoperiod. In order to gain an insight into the possible molecular mechanisms involved in this remarkable process, a number of key flowering-transition-associated genes were analyzed at different development stages that included the vegetative-to-reproductive growth transition. The gene expression analysis was based on a PPI network derived from the STRING database based on the *A. thaliana* genome, the results of which should referred to as tentative, considering that these mechanistic inferences are based on homologous genes and prior knowledge from *A. thaliana*. Thus, their functional conservation in *Ah* remains to be validated. As expected, a significant increase in *FT* expression in meristems and flowers was detected in R-BLL-exposed plants during the initiation (8 WAC) and fulfillment (12 WAC) of flower development. Moreover, the PCA1 analysis clearly suggested the relevance of high *FT* gene expression levels in the meristems and in flowers and, therefore, in the R-BLL-induced flowering in in vitro-cultured *Ah* plants maintained under a long-day photoperiod. Following in importance were the *LFY* and *CO* genes, whereas the *TFL1* repressor was strongly downregulated. The latter is in accordance with the notion that all flowering pathways necessarily converge at the florigen–anti-florigen hormone system that dictates, for example, that the balance between *FT* and *TFL1* will define the plant growth stage as indeterminate or determinate by modulating the formation of vegetative or reproductive tissues in the apical and axillary meristems [39]. The observed reduction in *TFL1* expression from 8 WAC onwards and in its varying relevance in the PCA were in accordance with its role as a key regulator of *FT* expression, and thereby of flower induction, which is defined through the 14-3-3 protein-mediated competition for the FD bZIP transcription factor (TF) [40]. The proposal that FT/TFL1 expression ratio may reflect a shift in floral developmental programming in *Ah*, based solely on transcriptional profiles, will require further experimentation, e.g., using protein-level or interaction assays, to be functionally validated.

Indirect support for the results of the present study can be drawn from a report describing that the overexpression of FT in *Brachypodium distachyon* led to early flowering and a determinate inflorescence structure in a day-length-independent manner [41]. Likewise, differences in the number and type of cis-acting elements in the promoter regions of PEBP genes such as *FT* have been proposed to define the plant’s perception of the photoperiod and its translation into long-day and short-day flowering habits [42]. Thus, contrary to short-day-flowering *Ah* plants, long-day-flowering *A. thaliana* lacks specific light-response elements, such as AE-Boxes and Gap-motifs [42]. In this context, the analysis of the cis-acting elements of the *Ah FT* gen promoter found twenty-one light-responsive cis-acting elements, including eleven box-4 (ATTAAT) motifs, four GT1-motifs, two TCT-motifs, two G-boxes, one I-box, and one GATA-motif. Among these, G-boxes [43], I-boxes [44], and GT1-motifs [45] are known to be regulated by red light, whereas GATA and I box motifs most likely mediated blue light sensitivity, similarly to previous findings reported in *A. thaliana* [46,47]. Moreover, box-4 cis-acting elements present in *Dof* family genes, also identified in closely related *Chenopodium quinoa* plants, were shown to be enhanced by both red and blue light [48]. An admission should be made at this point that although the presence of light-responsive cis-elements suggests potential regulation by light quality, experimental validation, e.g., by means of promoter–reporter studies, will be required to confirm their functionality in *Ah*. Nonetheless, this analysis provides an interesting preliminary framework to hypothesize potential gene regulatory functions in *Ah*, thereby opening up avenues for future experimental investigation to better understand the molecular mechanisms underlying flowering induction in this species.

In addition to *FT*, the proposed relevance of the *LFY* gene in the R-BLL-induced early and photoperiod-insensitive flowering in *Ah* was congruent with its role as a unique plant-specific TF involved in meristem floral fate and flower initiation [49]. Thus, in the present study, the induction of *LFY* gradually increased to reach the highest expression levels in the flowering stage, which were between ca. 9- and 50-fold higher than those detected in earlier vegetative and vegetative-to-reproductive transition stages. The chromatin-modifying capacity of LFY allows the binding of several other TFs to modulate several developmental processes, including flowering. It achieves the latter by acting downstream of FT to regulate floral organ development [50]. In this sense, it is tempting to speculate that its role in R-BLL-induced early and photoperiod-independent flowering could have involved the recruitment of a whole gamut of bZIP TFs and others that are known to regulate flowering time via changes in the structure of SWI/SNF-type complexes [51,52,53]. Also, the high levels of *LFY* expression detected in flowers of in vitro-R-BLL-treated *Ah* plants could have also contributed to promoting the accelerated flowering detected, similar to the early-flowering-related phenotypes observed in transgenic rice, tobacco, and hybrid aspen and citrus plants overexpressing the *LFY* gene [54,55,56,57]. Additionally, a transcriptome analysis of flower induction in ginger (*Zingiber officinale*) plants kept under red light indicated that, in coincidence with the present study, *LFY*, *CO*, and *FT* were highly upregulated and associated with flowering [58]. However, these comparisons should be taken with caution until more detailed evolutionary analyses and functional studies can clarify the conservation and divergence of these mechanisms, particularly in non-model species. In addition, similar gene expression patterns under red light in *Z. officinale*, an evolutionary divergent species, limit their direct applicability to *Ah*.

The additional importance allegedly assigned to *CO* in the early and photoperiod-independent flowering produced in in vitro *Ah* plantlets grown under R-BLL is supported by the role that this gene is known to play in the photoperiod-mediated control of flowering in other plant models. In *A. thaliana*, for example, *CO* is part of the regulatory hub in the photoperiodic flowering pathway that promotes long-day flowering by inducing *FT* expression in leaves [59]. Following the cautionary tone of this discussion, it must be acknowledged that although the evidence linking CO with the red–blue light induction of flowering under long-day conditions in *Ah* is compelling, its precise functional role in these plants and whether it operates similarly to short-day photoperiods remain to be determined.

Likewise, the PCA obtained during flowering revealed that the fourth most relevant gene coded for the PHL protein, which is known to suppress the degradation of CO by complexing with the PHYB sensor of red light [50]. Its projected importance in the accelerated flowering phenotype observed in *Ah* plants cultured under R-BLL and a long-day photoperiod is highlighted by the fact that *A. thaliana phl* mutants showed late-flowering phenotypes under long-day conditions, partly due to reduced *CO* levels [29]. In this sense, the importance of the CRY2 blue light sensor is derived from its positive regulation of CO. However, it remains to be determined how the known antagonistic action of *cry2* and *phyb* mutants observed toward flowering development in response to different light wavelengths [60] may have affected the accelerated flowering of *Ah* plants in response to R-BLL during long-day photoperiods. On the other hand, the reduced expression of the *SPA1* gene observed in the flowers of R-BLL-treated *Ah* plants could have affected its capacity to regulate the photoperiodic regulation of flowering time [61], thereby facilitating early flowering under long-day photoperiods in a plant with a short-day flowering habit. Also, despite the fact that PHYB suppresses the expression of flowering genes by promoting CO degradation, its positive expression, detected in the flowers of R-BLL-treated *Ah* plants, may have had a similar effect to that observed in *A thaliana PHYB*-overexpressing plants, which showed an early-flowering phenotype [62]. Additionally, the R-BLL exposure used to produce early-flowering *Ah* plants under long-day photoperiods led to induced levels of the *CRY2* gene in the periods corresponding to the initialization and culmination of flowering. This suggested that in response to the blue light component of the R-BLLs, the photoexcited CRY2 could have interacted with SPA1 or COP1 to suppress COP1 activity and, consequently, avoid CO degradation, thereby initiating the floral transition that, in the experimental conditions employed, could have been photoperiod-insensitive [63]. Conversely, the high-red-light conditions used in the present study could have promoted the binding of PHYB with SPA1, as proposed based on previous findings in *A. thaliana*, thereby preventing the formation of the COP1–SPA1 protein complex to enable the accumulation of photomorphogenesis-promoting proteins [64]. It should be emphasized, once more, that further experimental validation in *Ah* is required to certify the possible scenarios mentioned above.

Further, the PCA indicated the relevance, at 12 WAC, of *LEC1*, coding for a nuclear TF Y subunit B-9, known to be a transcriptional activator of genes required for both embryo maturation and cellular differentiation. Its importance at this stage probably contributed to the successful development of mature viable grain amaranth seeds with a high germination rate [65]. Similarly, the induction of the *NOB1* gene, coding for the NIN1 (ONE) BINDING PROTEIN1, in the 8 WAC pre-flowering stage in *Ah* grown under R-BLLs could be significant considering that this protein is essential for embryogenesis and pollen development processes required for adequate seed development [32]. In contrast, the downregulated expression of the ribosomal protein RPS18 gene detected at 12 WAC was suggestive of its minor influence on the flowering process in *Ah*, despite the fact that this protein is required for cell division [66] and is specifically expressed in meristems and in tissues with high cell division activity [67], whereas the *RPL21C* gene, coding for a large-subunit (60S) ribosomal protein, was induced predominantly in *Ah*’s stems, meristems, and flowers. The latter pattern could have been indicative of this gene’s key role in the maintenance of cellular activity in *Ah* plants, similarly to its participation in the regulation of chloroplast functions in *A. thaliana*. Additionally, it could have favored flowering in *Ah* due to its capacity to interact with epigenetic factors that regulate *FLC*, a major repressor of flowering [68]. Furthermore, it has been shown that it is essential for embryonic development [69]. On the other hand, the generally downregulated expression of the *GIGANTEA* gene observed in *Ah* plants suggested that its participation in the regulation of the flowering time via a CO-independent pathway that involves the involvement of miR172 and the further activation of photoperiodic flowering through the induction of *FT* was probably irrelevant in the context of the acceleration of flowering by R-BLL in *Ah* [70]. PRR1, known to play a critical role in the regulation of the circadian clock in close relation to PHYB activity, was highly ranked in component 2 of the PCA at 12 WAC, together with other genes expressed when flowering in *Ah* was repressed. Thereby, its strongly downregulated expression during active flowering could have favored early flowering considering that, under red light conditions, the expression of *PRR1* was found to decline and release *FT* from repression [24].

Another possible explanation for the accelerated flowering produced in *Ah* plants exposed to R-BLLs could be linked to the high irradiance emitted by the R-BLLs (i.e., 200 μmol m^−2^s^−1^), compared with WLLs, whose emissions reached only 94 μmol m^−2^s^−1^. This difference was consistent with findings indicating that increased irradiance and long-day conditions can accelerate flower induction in several commercially important ornamental plants [71]. Likewise, cis-elements like the G-box and GATA motif present in the *Ah FT* gene promoter could have represented a contributing flowering-habit-altering factor, considering that they were found to respond positively to continuous high-irradiance light in *A. thaliana* [47].

Based on the information discussed above, a hypothetical model based on homology with *A. thaliana* was constructed to propose the possible mechanisms by means of which flowering induction under a long-day photoperiod was accelerated in *Ah* plants kept under R-BLLs with a 2.2-fold-higher irradiance than WLL, in addition to much more intense red and blue peaks (Figure 6). This model predicts that under R-BLL conditions, the upregulation of *FT*, the master regulator of flowering, was greatly favored, as well as the induction of other light-sensitive flowering regulators such as *LFY*, *CO*, *CRY2*, *SPA1*, *PHYB*, *LEC1*, *MDN1*, *NOB1*, *RPL21*, and *COP1*. Moreover, successful flowering was accompanied by the production of mature and viable seeds. In contrast, the model proposes that under WLLs, flowering was repressed by the favoring of protein–protein interaction that inhibited CO accumulation and flower-promoting FT activity.

As is frequently the case with explanatory models, it must be recognized that the one above is based on speculative associations or suggests putative functions rather than demonstrate direct causality. Therefore, future functional validation experiments, such as gene knockouts, overexpression studies, or protein interaction assays, will be needed to conclusively determine the causal roles played by these genes in flowering regulation in *Ah*.

## 4. Materials and Methods

### 4.1. Plant Materials

Seeds of *Ah* cv. Revancha were collected from mature plants grown under greenhouse conditions for 90 days in the spring–summer of 2023. During this period, the maximum and minimum temperatures recorded in the greenhouse were 32 °C and 17 °C, respectively. The collected seeds were sterilized by placing them for 20 min in a desiccator containing 30 mL of sodium hypochlorite and 30 mL of hydrochloric acid 6 N. For in vitro culture, the seeds were dissected using longitudinal excisions that permitted the extraction of two symmetrical embryo halves.

### 4.2. Flowering Induction Conditions

Half-zygotic embryos of *Ah* were cultured in MS medium [72], complemented with either 3% or 5% sucrose and 3 or 5 g/L of Gelrite (GELZAN; Sigma-Aldrich, St. Louis, MO, USA), respectively. All media were set at a pH of 5.8. A total of fifty flasks were cultured with ten embryo halves per flask, half of which were placed under white LED lights (WLL; ECOFIT T8 LED lamps, Philips; Amsterdam, The Netherlands) with a photosynthetic photon flux (PPF) of 29.3 µmol/s (PPF-Blue [B]), 24.5 µmol/s (PPF-Red [R]) and a total irradiance of 94 µmol/m^2^ s. The rest were maintained under LED luminaries emitting light with a 3:1 R: B ratio (R-BLL; VIVIDGRO T8 lamps; Lighting Science Group Corporation, West Warwick, RI, USA). These emitted only two peaks in the visible spectrum: the largest, at the red light-end, had a PPF-R of 124.3 µmol/s, and the smaller, at the opposite, blue light end, had a PPF-B of 36.1 µmol/s. The lamps had 200 µmol/m^2^ s irradiance. All flasks were cultured inside a growth room maintained under long-day photoperiod conditions (16 h light/8 h darkness), with a light/dark temperature of 28/18 °C (for more information about the light quality, refer to Appendix A and Appendix A).

### 4.3. Interaction Analysis of Proteins Involved in Amaranth Flowering

A gene network analysis with 0.90 confidence was performed using STRING v11.5 [73] based on *A. thaliana* and using homologous *Ah* genes deposited in the Phytozome network [74]. STRING was used as a preliminary tool to identify potential candidate genes with known roles in flowering regulation in order to generate hypothesis-generating data.

Only those genes coding for *A. thaliana* homologs with protein sequence identities greater than 38% were selected, to increase the likelihood of conservation of both functional domains and likely biological roles. These included the photoreceptors of red *PHYB* (AT2G18790) and blue *CRY2* (AT1G04400) light; the photomorphogenesis regulators *COP1* (AT2G32950) and *SPA1* (AT2G46340); the flowering regulators *CO* (AT5G15840), *FT* (AT1G65480), *TFL1* (AT5G03840), *LFY* (AT5G61850), and *PHL* (AT5G29000); the circadian rhythms sensors *GI* (AT1G22770) and *TIMING OF CHLOROPHYLL A/B BINDING PROTEIN/PSEUDO RESPONSE REGULATOR1* (*TOC1/PRR1*) (AT5G61380); the ribosomal proteins *RPS18* (ATCG00650) and *RPL21C* (AT1G35680); the ribosome biogenesis regulators *NOB1* (AT5G41190) and *MDN1* (AT1G67120); and the seed and embryogenesis development regulator *LEC1* (AT1G72390). Homologous sequences in *Ah* with greater than 60% sequence similitude with *A. thaliana* proteins were considered for the design of primers used for the qPCR assays (see below).

### 4.4. Isolation of RNA and qPCR Analysis

The gene expression assay was performed with RNA isolates from plants cultured under at two light conditions (R-BLL and WLL) and sampled at three different development stages: 4, 8, and 12 weeks after the in vitro cultivation (WAC) of half-zygotic *Ah* embryos. Stems, leaves, dissected meristems, and flowers were collected, if applicable, at each sampling stage for subsequent RNA extraction. Total RNA from each different organ was isolated using Trizol (Invitrogen, Carlsbad, CA, USA); its purity and concentration were determined using a NanoDrop 2000 apparatus as instructed (ThermoFisher Scientific; Waltham, MA, USA), and its integrity was confirmed by 1% agarose gel electrophoresis.

Total RNA samples (1 μg) were reverse-transcribed to generate the first-strand cDNA as previously described [75]. The cDNA obtained was diluted 25-fold with deionized–distilled water prior to quantitative PCR (qPCR) analyses. These were performed using SYBR Green detection chemistry and a CFX96 Real-Time System (Bio-Rad, Hercules, CA, USA). The primers used are listed in Appendix A. The primers were designed using IDT Primer3 software (version 2.2.3; Integrated DNA Technologies, Inc., Coralville, IA, USA). The expression of the genes EF1 α (elongation factor 1 α) and β-TUBULIN was used as a reference for calculating the relative expression of the target genes under WLL and R-BLL using the 2^−∆∆CT^ method [35,76]. Log2-transformed gene expression values were subsequently employed to calculate the proportional changes in gene expression between WLL and R-BLL conditions.

### 4.5. Principal Component Analysis (PCA) of Gene Expression

All data derived from the quantitative gene expression assays under R-BLL and WLL conditions were subjected to a PCA after standardizing to unit variance. The resulting factor scores of PC1 and PC2 were tested in a two-way analysis of variance (ANOVA). Data analyses were carried out using Rstudio version 2024.09.0 + 375 [77].

### 4.6. Analysis of Cis-Acting Regulatory Elements in the Ah FT Gene

The Plant CARE database [78] was utilized to identify potential cis-acting regulatory elements within the promoter region of the *Ah FT* gene. The analysis spanned 2000 base pairs upstream of its transcription start site.

## 5. Conclusions

The results of this study present evidence suggesting that the coordinated action of CO, the integrator of the light signal into flowering, FT, the master regulator of flowering, and LFY, the controller of meristem fate and flower morphology, coupled with the much-reduced influence of the flowering repressor, TFL1, in meristems and in developing flowers, could have led to early flowering in *Ah* plants cultivated in vitro under R-BLLs and under a long-day photoperiod not conductive to reproductive development in these plants.

They also support the proposal that CRY2 and PHYB photoreceptors were probably important contributors via their possible inhibition of the formation of the COP1-SPA1 CO-repressing complex and subsequent blockage of *FT* induction. A high irradiance condition under R-BLLs compared with WLLs is also proposed as a contributing factor to the induction of early flowering in *Ah* plants cultivated in vitro and under a long-day photoperiod. Finally, increased protein synthesis mediated by the ribosome biogenesis factors NOB1 and MDN1 and the ribosomal protein RPL21 may have also contributed, not only to induce early flowering but also to ensure seed production in conjunction with LEC1.

The current protocol is being applied successfully in stable genetic transformation and genome-editing procedures of grain amaranths by particle bombardment and/or *Agrobacterium tumefaciens* mediation.

## Figures and Tables

**Figure 1 plants-14-03134-f001:**
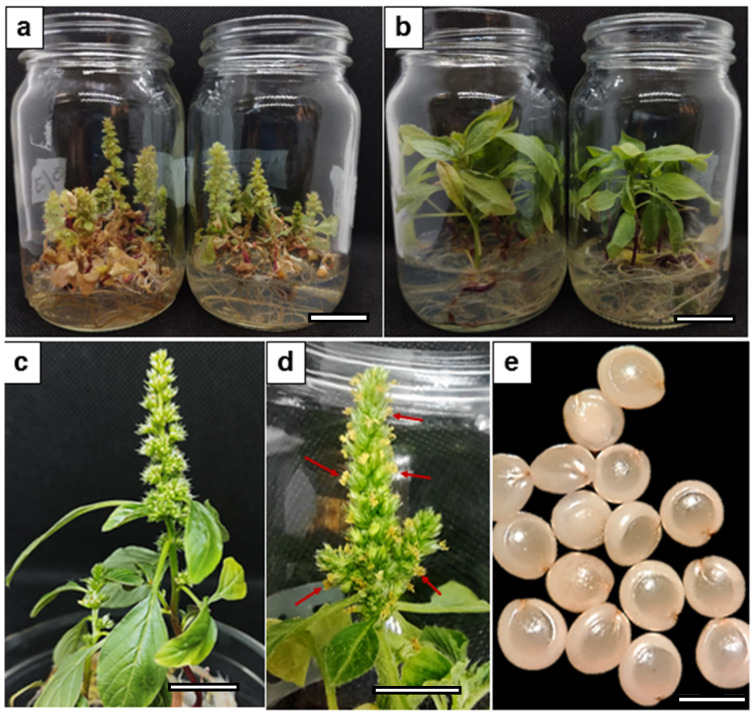
Aspects of induced flowering in *A. hypochondriacus* (*Ah*) plants grown under long-day photoperiods. (**a**) In vitro flowering of *Ah*, grown in MS medium with 3% sucrose and 3 g/l gelrite under red–blue LED light (R-BLL) illumination; (**b**) non-flowering *Ah* plants grown under white LED light (WLL) conditions; (**c**) monoecious *Ah* flowers; (**d**) pollen-producing *Ah* inflorescences (indicated by red arrows), and (**e**) mature seeds produced from *Ah* plants grown in vitro under R-BLL illumination. Measuring bars represent 1.5 cm in (**a**,**b**), 1 cm in (**c**,**d**), and 1 mm in (**e**).

**Figure 2 plants-14-03134-f002:**
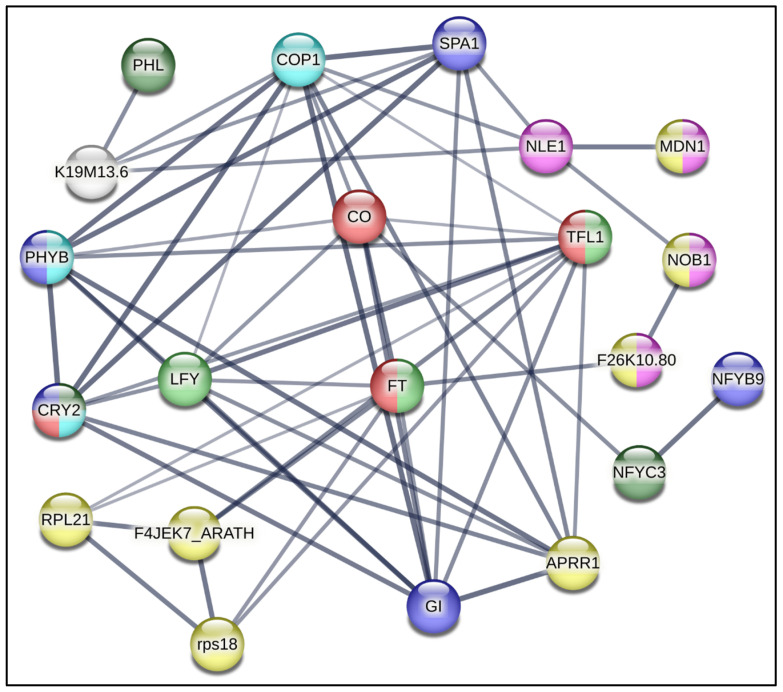
Flowering-regulating protein–protein interaction network. A STRING-based bioinformatic protein–protein interaction network relevant during the flowering process, at a 0.900 confidence level, was constructed using information garnered from the *A. thaliana* genome. It constituted the basis for the selection of homologous *Ah* genes for quantitative expression analysis. The genes chosen for analysis were *PHYB = PHYTOCHROME B* and *PHL = PHYTOCHROME-DEPENDENT LATE-FLOWERING*, which sense changes in red/far red light and transduce light photoreceptor phytochrome B signaling, respectively; *CRY2 = CRYPTOCHROME2*, which regulates photomorphogenic events through the detection of UV-A/blue light; *COP1 = CONSTITUTIVELY PHOTOMORPHOGENIC 1*; *SPA1 = SUPPRESSOR OF PHYA-1*; *CO = CONSTANS*, coordinately regulated by red and blue photoreceptors to determine photoperiodic flowering regulation; *FT = FLOWERING LOCUS T*; *TFL1 = TERMINAL FLOWER 1*; *LFY = LEAFY*; *GI = GIGANTEA*; *TIMING OF CHLOROPHYLL A/B BINDING PROTEIN/PSEUDO RESPONSE REGULATOR1* (*TOC1/PRR1*); *RPS18 = RIBOSOMAL PROTEIN S18*; *RPL21C = RIBSOMAL PROTEIN L21 SUBUNIT C*; *NOB1 = NIN1* (*ONE*) *BINDING PROTEIN 1*; *MDN1 = MIDASIN HOMOLOGUE 1*; and *LEC1 = LEAFY COTYLEDON1*.

**Figure 3 plants-14-03134-f003:**
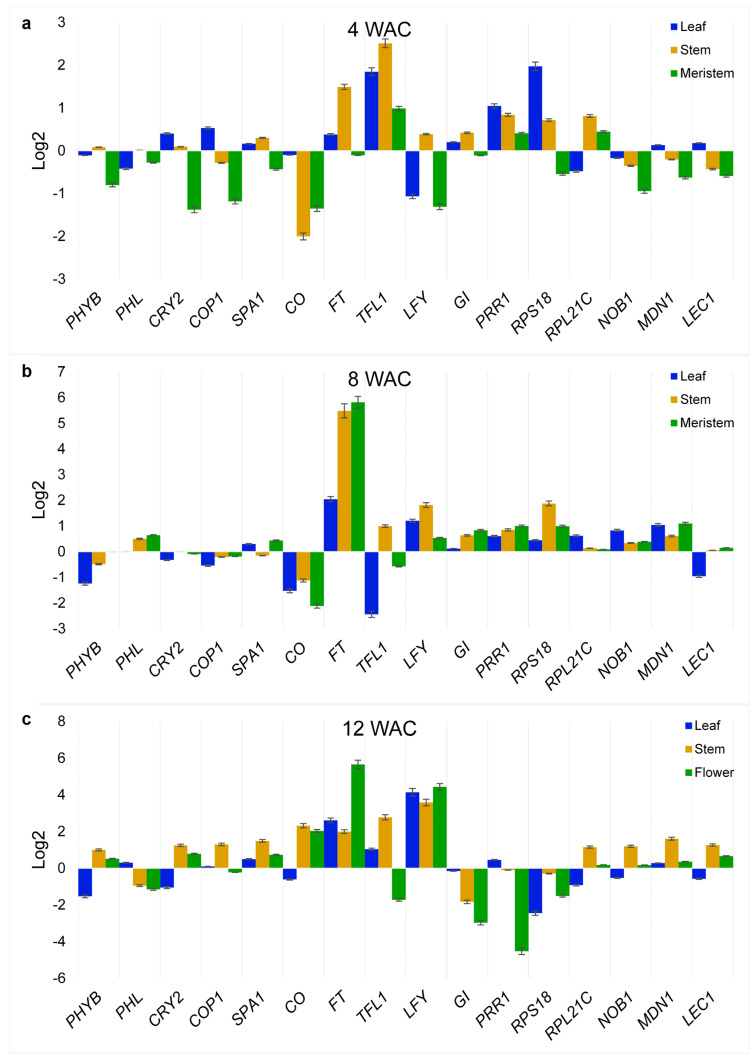
Time-course quantitative PCR analysis of the selected *A. hypochondriacus* (*Ah*) flowering-related genes. Expression levels of genes selected from the protein-protein interaction network shown in Figure 2 were determined in leaves, stems, shoot apical meristems (SAMs) and/or flowering meristems (FMs), and flowers at three different developmental stages in *Ah* plantlets cultivated in vitro either under red–blue (R-BLL) or white LED light (WLL) illumination and in the context of a long photoperiod. Sampling was performed at (**a**) 4 weeks, (**b**) 8 weeks, and (**c**) 12 weeks after cultivation of the *Ah* half-embryo explants on MS media. The relative expression values in each experimental condition were calculated according to Livak and Schmittgen [35] and normalized against the *ELONGATION FACTOR 1α* and *β-TUBULIN Ah* house-keeping genes. The results show the log2 of the ratio of relative expression levels detected under R-BLL and WLL, respectively. The genes analyzed were *PHYB*, *PHL*, *CRY2*, *COP1*, *SPA1*, *CO*, *FT*, *TFL1*, *LFY*, *GI*, *TOC1*/*PRR1*, *RPS18*, *RPL21C*, *NOB1 = NIN1*, *MDN1*, and *LEC1* (for the meaning of the gene abbreviations, refer to Figure 2). The results represent at least three technical replicates of a biological experiment that was repeated at least thrice with similar results.

**Figure 4 plants-14-03134-f004:**
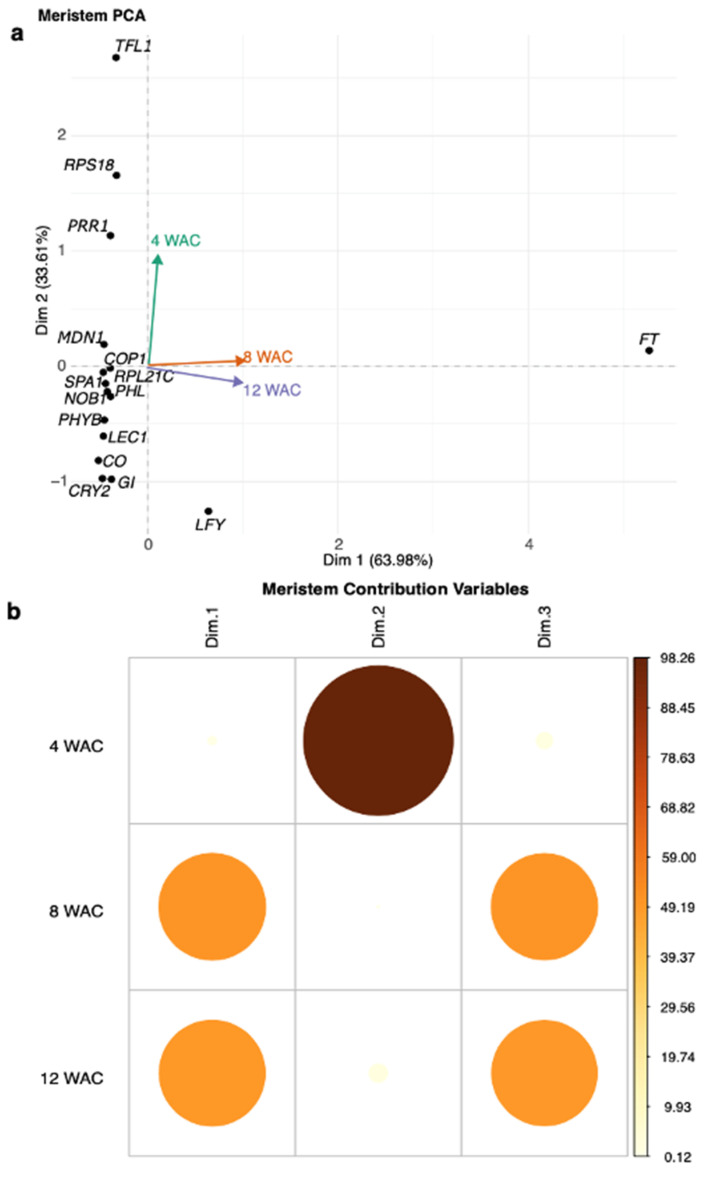
Principal component analysis (PCA) of the quantitative gene expression assays detected in apical meristems and flowers. In (**a**) the PCA1 (*x*-axis, increasing average expression) indicates that 63.98% of the genes analyzed had significantly modified levels of expression. Here, the *FT* gene exhibited the higher gene contribution in gene expression (i.e., 90.41), making it, by far, the most relevant gene in flowering development under R-BLL conditions 12 WAC. PCA2 (*y*-axis, increasing positive trend) explained 33.61% of the variance. It indicated that the most relevant contribution to this component during flowering at 12 WAC was from *TFL1* (44.46). It was followed by *RPL18* (17.00), *LFY* (9.8), and *PRR1* (7.97). (**b**) Components 1 and 3 were enriched in genes whose maximum levels were reached at 8 and 12 WAC, while component 2 grouped those genes whose expression was predominant at 4 WAC, representing the pre-flowering vegetative stage.

**Figure 5 plants-14-03134-f005:**
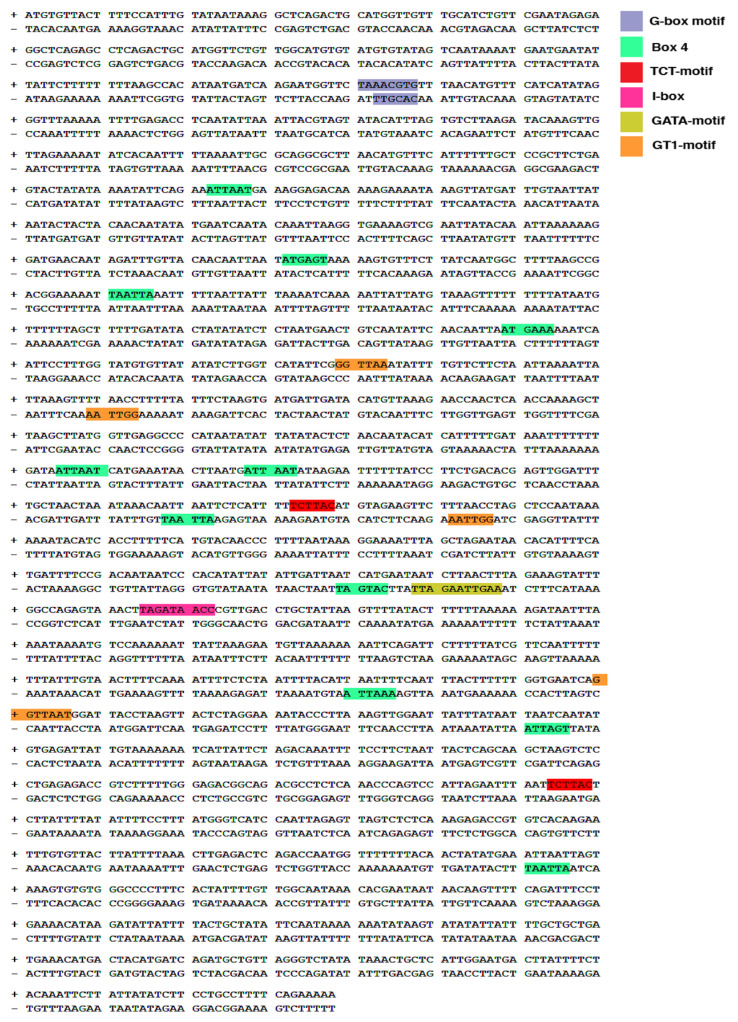
Light-responsive elements (LREs) present in the promoter region of the *A. hypochondriacus FT* gene. The figure shows 2000 nucleotides upstream of the 5’-UTR of the *Ah FT* gene. Nucleotide sequences representing potential LREs are highlighted in different colors, representing Box4 motifs (green boxes); GT1 motifs, (orange boxes); G-Box motifs, (blue boxes); TCT motifs, (red boxes); GATA motif (mustard-yellow box); and I-box motif (pink box). The “+” and “−” symbols indicate base pairs downstream and upstream of the transcription start site, respectively.

**Figure 6 plants-14-03134-f006:**
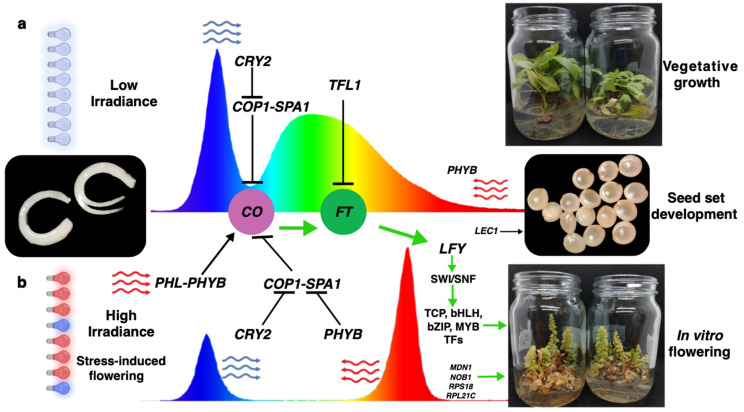
Induced flowering in in vitro-cultured *A. hypochondriacus* plantlets maintained under red–blue LED lights in the context of a non-flowering-inducing long-day photoperiod: proposed model based on homology with *A. thaliana*. Two hypothetical scenarios are shown: (**a**) half-zygotic embryos cultivated during long-day photoperiods under white LED light (WLL) luminaries with low irradiance; the CRY2 blue light receptor becomes the dominant regulator. This triggers the day-time repression of CO that contributes to blocking the formation of the SPA1-COP1 complex. Furthermore, under these conditions, the TFL-mediated repression of flowering via FT is favored. (**b**) In vitro cultivation maintained under 3:1 red–blue (R-B) LED luminaries with high irradiance activates a different flower-inducing process that involves the formation of a PHL-PHYB complex that favors the activation of CO; this step is further promoted by the inhibited formation of the SPA1-COP1 complex by the PHYB red light receptor and CRY2. The sum of these events promotes the activation of FT, which leads to the downstream accumulation of LFY that allows the chromatin opening of the SWI/SNF complex and the binding of transcription factors (TFs) that stimulate flower development. At the same time, ribosomal biogenesis via RPS18 and other proteins aids in the formation of in vitro flowers capable of generating fertile seeds via LEC1.

**Table 1 plants-14-03134-t001:** Quantitative parameters obtained from the accelerated flowering of *A. hypochondriacus* plantlets cultured in vitro under 3:1 red–blue LED light luminaries in the context of a long-day photoperiod.

N° Experiment	N° Cultured Embryos	N° Surviving Plants	Flowering Plants (%)	N° Seeds/Plant	Germination (%)
1	100	84	74.8	163	68.3
2	100	88	65.5	136	74.5
3	100	98	75.1	122	74.5
4	100	86	68.5	146	67.2
5	100	94	72.4	131	65.8
Mean value		90	71.3	139.6	70.0

## Data Availability

Data are contained within the article and Appendix A.

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
