# Peer review of "High Red–Blue Light Ratio Promotes Accelerated In Vitro Flowering and Seed-Set Development in Amaranthus hypochondriacus Under a Long-Day Photoperiod"

_plants, 2025, doi:10.3390/plants14203134_

Round 1
Reviewer 1 Report
Comments and Suggestions for Authors
I have reviewed a manuscript titled „High red: blue light ratio promotes accelerated in vitro flowering 2 and seed-set development in Amaranthus hypochondriacus un-3 der a long-day photoperiod “ by authors Bermudez-Valle et al.. The authors developed an efficient in vitro protocol for Amaranthus hypochondriacus by manipulating light conditions, which successfully induced early flowering, viable seed production, and revealed key molecular regulators underlying the process.
The paper is well-written, and I have no remarks.
One technical mistake that I find is:
Figure 3 - Since the paper refers to Figure 3a (Line155) , and it is also mentioned in its description, labels a, b, and c should be added next to the figure.
Line 156 - .. and RPS18 and were among the genes most highly induced in leaves and..- red and is redundant
Author Response
I have reviewed a manuscript titled „High red: blue light ratio promotes accelerated in vitro flowering 2 and seed-set development in Amaranthus hypochondriacus un-3 der a long-day photoperiod “ by authors Bermudez-Valle et al.. The authors developed an efficient in vitro protocol for Amaranthus hypochondriacus by manipulating light conditions, which successfully induced early flowering, viable seed production, and revealed key molecular regulators underlying the process.
The paper is well-written, and I have no remarks.
One technical mistake that I find is:
Figure 3 - Since the paper refers to Figure 3a (Line155), and it is also mentioned in its description, labels a, b, and c should be added next to the figure.
We are sorry for this typographical error. The letter labels somehow got lost during the figure´s transfer to the main manuscript. Figure 3 has now been amended to show the appropriate labeling.
Line 156 - .. and RPS18 and were among the genes most highly induced in leaves and..- red and is redundant
The duplicated “and” has been deleted from this sentence. However, we failed to understand what you meant by “…red and is redundant”.
Reviewer 2 Report
Comments and Suggestions for Authors
The research showed the induction of flowering in Ah plants grown in vitro using 3:1 red: blue LED luminaries (R-BLL) under a long-day photoperiod. The plants developed rapidly and successfully completed their reproductive cycle, generating flowers and viable seeds that germinated normally. In this research, STRING database v12.0 network was queried to select the genes whose expression patterns could provide an understanding of the possible molecular mechanism involved in the uncharacteristic flowering process here described. This information was integrated into a model describing the role played by R-BLL illumination in the induction of early flowering in Amaratnus plants cultured in vitro under a non-flowering inducting long-day photoperiod. The probable mechanisms by means of which flowering induction under a long-day photo period was accelerated in Ah plants was proposed. What about future research? Not only the application. The conclusion could be more clearly written, because of a lot of detailed information.
You should decribe the hypothese of this research in Introduction.
Methods are properly and sufficiently described.
The seeds are viable. But what about germination?
Fig. 2 I can't see the light influence. There are the results for both together?
Author Response
The research showed the induction of flowering in Ah plants grown in vitro using 3:1 red: blue LED luminaries (R-BLL) under a long-day photoperiod. The plants developed rapidly and successfully completed their reproductive cycle, generating flowers and viable seeds that germinated normally. In this research, STRING database v12.0 network was queried to select the genes whose expression patterns could provide an understanding of the possible molecular mechanism involved in the uncharacteristic flowering process here described. This information was integrated into a model describing the role played by R-BLL illumination in the induction of early flowering in Amaratnus plants cultured in vitro under a non-flowering inducting long-day photoperiod. The probable mechanisms by means of which flowering induction under a long-day photo period was accelerated in Ah plants was proposed. What about future research? Not only the application. The conclusion could be more clearly written, because of a lot of detailed information.
You should describe the hypothesis of this research in Introduction.
A working hypothesis of this study was incorporated in lines 99-102 of the revised manuscript.
Methods are properly and sufficiently described.
The seeds are viable. But what about germination?
Perhaps you missed part of the data presented in Table 1, where the germination rate of the seeds produced by Ah plants that underwent accelerated flowering hovered between ca. 65 and 75%.
Fig. 2 I can't see the light influence. There are the results for both together?
We are sorry, but it was not clear to us what you meant by this comment. We think that perhaps the confusion arose from the fact that no indication was given of which proteins are known to act as receptors of different light qualities in the protein network shown in Figure 2. Therefore, additional information in this respect was added in this figure legend in effort to correct this source of confusion.
Perhaps you meant Fig. 4 instead of Fig. 2? If that was case, please let us know so that appropriate corrections can be made.
Reviewer 3 Report
Comments and Suggestions for Authors
The MS has following flaws;
Over-reliance on Arabidopsis analogies without experimental support in Amaranthus.
Correlation = causation problem: gene expression increases are interpreted as causal.
In silico motif predictions used as major evidence, no functional assays.
Speculative mechanistic pathways (e.g., CO/FT/PHYB/CRY2 interplay) presented as fact.
Weak species comparisons (ginger, rice, Arabidopsis) without discussing evolutionary divergence.
Lines 47–49: You mention a "deficient integration of these plants into large-scale agricultural production" mainly due to mechanization issues. Could you provide stronger evidence or references that specifically link mechanization challenges with the limited adoption of grain amaranths?
Lines 78–81: The CO–FT module is described here based on Arabidopsis and other long-day plants. Given that amaranths are primarily short-day crops, is it valid to extrapolate this model directly? Are there reports of FT homologs characterized in Amaranthus spp.?
LInes 238–243: You state molecular mechanisms of in vitro flowering are "poorly studied." Could you support this with more recent references or clarify what is specifically lacking?
Why was the STRING database (based on Arabidopsis) chosen given the evolutionary distance from Amaranthus? Could this limit the reliability of the gene network predictions?
Line 256-264 - Over-interpretation of expression data without functional validation.
You claim the FT/TFL1 balance defines growth stage — but did you experimentally test TFL1 protein interactions or only infer from mRNA levels?
Lines 268–277: Reliance on motif prediction without experimental confirmation weakens the conclusion. The cis-element analysis is presented. Were these motifs experimentally validated (e.g., promoter::reporter assays), or are they purely in silico predictions?
Lines 297–298: You compare to ginger transcriptome data — but is this valid given the monocot–dicot divergence? Comparisons across distantly related taxa may not be robust.
Lines 299–307: How do you differentiate CO’s role in Amaranthus (a short-day plant) from its canonical role in long-day Arabidopsis? Assumption of conserved function without experimental verification in the studied species.
Line 327-329; You suggest PHYB–SPA1 interaction prevents COP1–SPA1 formation. Was this interaction demonstrated in Amaranthus or simply inferred from Arabidopsis? Mechanistic inferences entirely speculative without experimental evidence.
Author Response
We thank you for your critical and constructive comments, which have helped improve the clarity, rigor, and scientific precision of the manuscript. Below, we provide a detailed point-by-point response. Modifications made to the manuscript are indicated where applicable.
The MS has following flaws;
Over-reliance on Arabidopsis analogies without experimental support in Amaranthus.
This comment is appreciated. Indeed, several hypotheses presented in the manuscript rely on conserved pathways described in Arabidopsis. This was necessary due to the limited availability of genomic or functional data in Amaranthus. However, the manuscript has been modified to clearly state that these mechanistic inferences are based on homologous genes and prior knowledge from Arabidopsis, and that their functional conservation in Amaranthus remains to be validated. Phrases such as “suggests a role,” “may be involved,” and “putative regulatory mechanism” have replaced more definitive statements. We also now acknowledge this limitation explicitly in both the discussion and conclusions.
Furthermore, we used this analogy based on the percentage identity of protein sequences between Amaranthus and Arabidopsis. In all cases, only homologs with protein sequence identities greater than 38% were included, demonstrating conservation of both functional domains and, likely, biological roles. Additionally, the protein-protein interaction (PPI) networks were constructed using a high STRING confidence threshold (0.9), which includes only experimentally validated and/or strongly predicted interactions conserved across multiple species. Thus, the resulting network reflects not only Arabidopsis-based predictions but also cross-species conserved interactions supported by experimental and computational evidence. This increases the reliability of our inferences, although we acknowledge that species-specific validation remains necessary.
Correlation = causation problem: gene expression increases are interpreted as causal.
We acknowledge the reviewer’s observation regarding the distinction between correlation and causation. We fully acknowledge that increased gene expression alone does not establish a causal role in flowering induction. In our study, gene expression data combined with STRING-based protein interaction networks served to generate an hypothesis about potential regulatory pathways involved in R-BLL-induced flowering in Amaranthus hypochondriacus (Ah). However, we have revised the manuscript to clearly state that these findings indicate associations or suggest putative functions rather than demonstrate direct causality. We emphasize the need for future functional validation experiments, such as gene knockouts, overexpression studies, or protein interaction assays, to conclusively determine the causal roles of these genes in flowering regulation in Ah. This limitation is now explicitly discussed in the manuscript.
We have also taken steps to revise the language throughout the manuscript to emphasize that our expression analyses are strictly correlative. We now clarify that no functional validation (e.g., gene silencing, overexpression, or mutant analysis) was conducted. We also discuss the need for future functional analyses to validate these correlations.
In silico motif predictions used as major evidence, no functional assays.
We agree. The cis-regulatory motif analysis was conducted entirely in silico using PLANTCARE, and no promoter-reporter or functional binding assays were performed. The revised manuscript now makes this limitation explicit and presents the motif data as a set of putative regulatory elements rather than confirmed functional sequences. We have now included the following statement (refer to lines 310-316): “Admission should be made that although the presence of light-responsive cis-elements suggests potential regulation by light quality, experimental validation, e.g, by means of promoter-reporter studies, is still required to confirm their functionality in Ah. Nonetheless, this analysis provides an interesting preliminary framework to hypothesize potential gene regulatory functions in Ah, thereby opening avenues for future experimental investigation to better understand the molecular mechanisms underlying flowering induction in this species”.
Speculative mechanistic pathways (e.g., CO/FT/PHYB/CRY2 interplay) presented as fact.
This is a valid point. We have carefully revised the discussion and Figure 6 legend to describe these interactions as hypothetical and derived from Arabidopsis. For instance, the model in Figure 6 is now described as a “proposed model” and includes qualifying statements such as “based on homology with Arabidopsis” and “hypothetical scenarios.” We emphasize that the roles of CO, PHYB, and CRY2 in Ah remain speculative and require empirical confirmation.
Weak species comparisons (ginger, rice, Arabidopsis) without discussing evolutionary divergence.
We appreciate the comment regarding comparisons among evolutionarily distant species such as ginger, rice, and Arabidopsis. We acknowledge that phylogenetic differences may limit direct extrapolation of molecular mechanisms across species. However, given the lack of detailed molecular information for Ah, we used these models as initial references to infer potentially conserved pathways involved in floral induction under specific conditions. Importantly, these functional inferences were based on protein sequence identity and the conservation of key functional domains between Ah and model species, providing a rationale for proposing these hypotheses. We have emphasized in the manuscript that such comparisons are made cautiously and that the proposed pathways remain hypothetical until experimentally validated in Ah. Future work will focus on more detailed evolutionary analyses and functional studies to clarify the conservation and divergence of these mechanisms in non-model species.
Lines 47–49: You mention a "deficient integration of these plants into large-scale agricultural production" mainly due to mechanization issues. Could you provide stronger evidence or references that specifically link mechanization challenges with the limited adoption of grain amaranths?
Thank you for your comment. WE have added the following sentence to support these arguments (refer to lines 50-54): “This limitation is caused by several agronomic characteristics, such as small seed size, irregular seed maturation, and plant architecture, which complicate conventional mechanized harvesting and processing. Additionally, the lack of suitable equipment adapted for grain amaranth cultivation further restricts mechanized production”.
These factors are well documented in the literature and contribute significantly to the low integration of grain amaranths in commercial agriculture. To overcome these issues, breeding and biotechnological approaches, including genetic transformation and genome editing, are being explored to develop varieties better suited for mechanized agriculture.
Lines 78–81: The CO–FT module is described here based on Arabidopsis and other long-day plants. Given that amaranths are primarily short-day crops, is it valid to extrapolate this model directly? Are there reports of FT homologs characterized in Amaranthus spp.?
Valid comment. We recognize that Ah is a short-day species, while Arabidopsis is a long-day flowering plant, and therefore the canonical CO-FT module described in Arabidopsis may not fully apply. However, core components of the photoperiod pathway such as CO and FT are conserved across many species, making the CO-FT module a useful framework to study flowering regulation in Amaranthus.
We have revised the manuscript to explicitly acknowledge this limitation in the following paragraph (refer to lines 102-106): “…in which the observed increase in CO and FT transcripts under red-blue LED conditions in a long-day photoperiod suggested a possible alternative or modified flowering regulatory mechanism in Ah, distinct from the classical A. thaliana model. Still, these pathways remain hypothetical and require experimental validation to be validated”.
LInes 238–243: You state molecular mechanisms of in vitro flowering are "poorly studied." Could you support this with more recent references or clarify what is specifically lacking?
We have clarified that while in vitro flowering has been reported in over 100 species (as per [10]), studies that explore the molecular mechanisms, particularly under controlled light regimes and in underutilized crops like Ah, remain scarce. We now include two recent references, i.e. [36, 37], on in vitro flowering in other crops and in amaranth and quinoa, as well, to justify this statement (refer to lines 264-267).
Why was the STRING database (based on Arabidopsis) chosen given the evolutionary distance from Amaranthus? Could this limit the reliability of the gene network predictions?
We acknowledge that the STRING database is based on Arabidopsis and that this limits the predictive accuracy of interaction networks in non-model species like Ah. We now explain in the methods and discussion section that STRING was used only as a preliminary tool to identify potential candidate genes with known roles in flowering regulation. The predictions are now framed as hypothesis-generating rather than conclusive, and we emphasize the need for species-specific data to validate these interactions.
Furthermore, we used this analogy based on the percentage identity of protein sequences between Amaranthus and Arabidopsis. In all cases, only homologs with protein sequence identities greater than 38% were included, demonstrating conservation of both functional domains and, likely, biological roles. Additionally, the protein-protein interaction (PPI) networks were constructed using a high STRING confidence threshold (0.9), which includes only experimentally validated and/or strongly predicted interactions conserved across multiple species. Thus, the resulting network reflects not only Arabidopsis-based predictions but also cross-species conserved interactions supported by experimental and computational evidence. This increases the reliability of our inferences, although we acknowledge that species-specific validation remains necessary (refer to lines 274-277, 429-434 and 480-484).
Line 256-264 - Over-interpretation of expression data without functional validation.
You claim the FT/TFL1 balance defines growth stage — but did you experimentally test TFL1 protein interactions or only infer from mRNA levels? Were TFL1–FD protein interactions tested, or are they inferred?
We agree and have revised the manuscript to avoid over-interpretation. We now refer to the observed FT/TFL1 expression patterns as suggestive of a shift in meristem fate, and not as definitive evidence of a growth stage transition. Also, no protein-protein interactions were tested in this study. All references to TFL1–FD competition are based on known mechanisms in Arabidopsis and are now clearly labeled as inferred. This is mentioned in the revised MS as follows (refer to lines 291-294): “The proposal that FT/TFL1 expression ratio may reflect a shift in floral developmental programming in Ah, based solely on transcriptional profiles, will require further experimentation, e.g., using protein-level or interaction assays, to be functionally validated”.
Lines 268–277: Reliance on motif prediction without experimental confirmation weakens the conclusion. The cis-element analysis is presented. Were these motifs experimentally validated (e.g., promoter::reporter assays), or are they purely in silico predictions?
The answer is no. The motifs identified in the AhFT promoter were predicted using in silico tools and not validated experimentally. This is now clearly stated in the text, and the conclusions related to these motifs have been revised accordingly. We now emphasize the need for functional assays (e.g., promoter-reporter studies) to confirm their regulatory roles (refer to lines 310-316).
Lines 297–298: You compare to ginger transcriptome data — but is this valid given the monocot–dicot divergence? Comparisons across distantly related taxa may not be robust.
We acknowledge the large phylogenetic gap between monocots and dicots. We have revised the text to indicate that this comparison is merely illustrative and does not imply functional conservation. The sentence now reads (refer to lines 335-339): “However, these comparisons should be taken cautiously until more detailed evolutionary analyses and functional studies clarify the conservation and divergence of these mechanisms, particularly in non-model species. Moreover, similar gene expression patterns under red light in Z. officinale, an evolutionary divergent species, limit their direct applicability to Ah”.
Lines 299–307: How do you differentiate CO’s role in Amaranthus (a short-day plant) from its canonical role in long-day Arabidopsis? Assumption of conserved function without experimental verification in the studied species.
We agree and have revised this section accordingly (refer to lines 345-349). “Following the cautionary tone of this discussion, it must be acknowledged that although the evidence linking CO with the red-blue light induction of flowering under long-day conditions in Ah is compelling, knowledge of its precise functional role in these plants and whether it operates similarly to short-day photoperiods, remains to be determined”.
Line 327-329; You suggest PHYB–SPA1 interaction prevents COP1–SPA1 formation. Was this interaction demonstrated in Amaranthus or simply inferred from Arabidopsis? Mechanistic inferences entirely speculative without experimental evidence.
The answer is no, again. This interaction was inferred from Arabidopsis data. The manuscript now clearly states (refer to lines 374-379): “Conversely, the high red-light conditions used in the present study could have promoted the binding of PHYB with SPA1, as proposed on previous findings in A. thaliana, thereby preventing the formation of COP1–SPA1 protein complex to enable the accumulation of photomorphogenesis-promoting proteins [63]. Emphasis should be made, once more, that further experimental validation in Ah is required to certify the possible scenarios men-tioned above”.
Reviewer 4 Report
Comments and Suggestions for Authors
This paper focuses on the study of the manipulation of in vitro flowering in the grain amaranth's regeneration recalcitrance of Amaranthus hypochondriacus, and analyzes the basic molecular mechanisms, gene expression patterns, and key flowering related genes of the observed phenomena. Ultimately, it concludes that model is proposed to explain how this light-regulated molecular framework enabled in vitro flowering and seed production in Ah plants kept under long-day photoperiods. This study has both significant practical and theoretical value in production, providing a useful supplement to the research on light controlled development theory.
Through review, it was found that the article is scientifically designed, with sufficient discussion basis, rigorous narrative logic, and significant conclusions. The article has strong innovation and overall high level.
There are two small questions that need to be confirmed:
1. What does the white horizontal bar in the small picture in Figure 1 represent?
2. There are relatively few references in the past three years, which can be updated or supplemented appropriately.
Author Response
This paper focuses on the study of the manipulation of in vitro flowering in the grain amaranth's regeneration recalcitrance of Amaranthus hypochondriacus, and analyzes the basic molecular mechanisms, gene expression patterns, and key flowering related genes of the observed phenomena. Ultimately, it concludes that model is proposed to explain how this light-regulated molecular framework enabled in vitro flowering and seed production in Ah plants kept under long-day photoperiods. This study has both significant practical and theoretical value in production, providing a useful supplement to the research on light controlled development theory.
Through review, it was found that the article is scientifically designed, with sufficient discussion basis, rigorous narrative logic, and significant conclusions. The article has strong innovation and overall high level.
There are two small questions that need to be confirmed:
- What does the white horizontal bar in the small picture in Figure 1 represent?
The horizontal bars represent “measuring bars” as mentioned at the bottom the figure´s legend.
- There are relatively few references in the past three years, which can be updated or supplemented appropriately.
Recent citations encompassing the years 2022-2025 were added to the revised MS. Please refer to citations 20-24 and 37.
Round 2
Reviewer 2 Report
Comments and Suggestions for Authors
The authors have corrected the work in the indicated aspects, and this manuscript is now ready for publication. The hypotheses were added, and the discussion was supplemented. The introduction is sufficiently described, and a few additional works have been added. The discussion is supplemented with a few important sentences. However, I conclude that the Authors feel the research requires further work and experimentation.
Reviewer 3 Report
Comments and Suggestions for Authors
The author has revised the MS as per the suggestion.